# Methylated Biochemical Fulvic Acid-Derived Hydrogels with Improved Swelling Behavior and Water Retention Capacity

**DOI:** 10.3390/ma17061448

**Published:** 2024-03-21

**Authors:** Chunhui Shi, Xifeng Lv, Jingfan Peng, Jikui Zhu, Fengqin Tang, Libing Hu

**Affiliations:** College of Chemistry and Chemical Engineering, Tarim University, Aral 843300, China; 120200030@taru.edu.cn (C.S.); 120120022@taru.edu.cn (X.L.); 7111221231@stumail.taru.edu.cn (J.P.); jkzhu@taru.edu.cn (J.Z.)

**Keywords:** humic acid, hydrogel, swelling rate, water-retaining agent

## Abstract

Although humic acids (HAs) have been used to prepare absorbent hydrogels, their applications in many areas, such as agriculture, wastewater treatment and hygienic products, are not satisfactory due to their low solubility in organic solvents. In this work, biochemical fulvic acid (BFA), as a kind of HA, was initially methylated for preparation of the methylated BFA (M-BFA), which contributed to enhancing the solubility in organic solvents. Then, M-BFA reacted with N,N′-methylene **diacrylamide** (MBA) in the N,N-Dimethylacrylamide (DMAA) solution, and the expected hydrogel (M-BFA/DMAA) was successfully obtained. XPS confirmed that there were more C=O and C-N groups in M-BFA/DMAA than in DMAA; thus, M-BFA/DMAA was able to offer more reactive sites for the water adsorption process than DMAA. The combined results of BET and SEM further demonstrated that M-BFA/DMAA possessed a larger BET surface area, a larger pore volume and a more porous structure, which were favorable for the transfer of water and accessibility of water to active sites, facilitating water adsorption and storage. In addition, the swelling ratio and water retention were investigated in deionized (DI) water at different conditions, including test times, temperatures and pHs. Amazingly, the swelling ratio of M-BFA/DMAA was 10% higher than that of DMAA with the water retention time from 100 to 1500 min. Although M-BFA/DMAA and DMAA had similar temperature sensitivities, the pH sensitivity of M-BFA/DMAA was 0.9 higher than that of DMAA. The results proved that M-BFA/DMAA delivered superior water retention when compared to the pristine DMAA. Therefore, the resultant materials are expected to be efficient absorbent materials that can be widely used in water-deficient regions.

## 1. Introduction

In the past decades, water scarcity has become one of the foremost challenges for human beings with the rapid development of society and increasing population growth. In particular, this phenomenon is terribly serious in water-deficient regions, which has seriously affected the agricultural development and human living environment in various countries [1,2,3]. Various strategies, such as seawater desalination, atmospheric water harvesting (AWH) and water-retaining agents, are considered as the most universal strategies for freshwater production. Although seawater desalination is a relatively mature technology, it is seriously restricted due to its limited suitability for coastal areas, so it is not applicable to inland water shortage areas far from the coast [4]. In addition, AWH, as an emerging technology, can be employed in all regions and is not hindered by time and space [5]. However, the low adsorption capacity and circulation rate of AWH materials heavily limit the final water production. Typically, lightly crosslinked hydrophilic polymers, as water-retaining agents, are able to absorb natural water through the network structure of hydrophilic groups, expand into the corresponding hydrogels after absorbing water, and then slowly release the water to the soil; this process is not restricted by time and space, nor does it consume energy [6,7]. Therefore, the development of suitable water-retaining agents for water storage is paramountly significant for crop harvest and human living.

So far, the majority of the important water-retaining agents have been developed to save water and fertilizer. As widely reported, poly-acrylic acid (PAA) salt [8], denatured starch [9,10] and cyanobacterial polysaccharides [11,12] have been used as raw materials for preparing water-retaining agents to alleviate drought stress. Aside from the water-retaining benefits, the reduced effect of the degradation products of the utilized chemicals on the environment and plant growth is also a critical issue [13]. For instance, PAA degradation is difficult in the natural environment; therefore, the large-scale application of PAA salt would result in a great environmental risk [14,15]. Furthermore, denatured starch has a low water absorption rate; thus, it is inevitable that a large quantity of denatured starch is necessary for water retention [16]. Accordingly, the employment of raw materials with the properties of compatibility, biodegradability and water adsorption to develop water-retaining agents is greatly significant.

Humic acids (HAs), as a natural product, are derived from the decay of both plants and animals under decomposition caused by microorganisms and moist environment conditions [17,18]. Consequently, HAs are abundant in nature, low-cost and easily biodegradable. On the other hand, HAs contain multifunctional aromatic constituents and aliphatic components and have a number of functional hydrophilic groups like phenolic hydroxyls and carboxylates, which are favorable for the preparation of water-retaining materials due to their water adsorption capacities [19,20,21]. For instance, hydrogels were prepared with the precipitation of HAs dissolved in an alkaline solution with washing and centrifugation by using a strong acid. Typically, irreversible hydrogels were obtained through physical bonds and contained 80–90 wt% water [22]. Unfortunately, the practical application of water-retaining materials prepared with HAs as raw materials is severely limited due to their low solubility in organic solvents [23,24].

Herein, biochemical fulvic acid (BFA) as a kind of HA was initially methylated to obtain M-BFA. During the following process, M-BFA reacted with N,N′-methylene diacrylamide (MBA) in N,N-Dimethylacrylamide (DMAA) solution with the help of ammonium persulfate ((NH_4_)_2_S_2_O_4_) as an active agent, and the expected hydrogel (M-BFA/DMAA) could be directly obtained via polymerization in the solution. In order to understand its ability to absorb and store water, the swelling properties of the obtained hydrogel were investigated at different times, temperatures and pHs.

## 2. Experiment

### 2.1. Materials

The materials used were N,N-Dimethylformamide (DMF), N,N-Dimethylacrylamide (DMAA), N,N′-methylene diacrylamide (MBA), sodium hydroxide (NaOH), hydrochloric acid (HCl, 98%), dimethyl carbonate (DMC), potassium carbonate (K_2_CO_3_), toluene (98%) and ammonium persulfate ((NH_4_)_2_S_2_O_4_). The above materials were all high-grade pure and purchased from Aladdin Reagent Co., Ltd, Shanghai, China.

### 2.2. Preparation of BFA

In order to obtain BFA, cotton straws (CSs) were first collected in the fields near Tarim University and then pulverized into powder by a pulverizer. Subsequently, the obtained powder was screened through 80-mesh screens to gain the desired powder with a particle size of ~150 μm, and the obtained powder was washed with deionized water three times to remove the impurities.

Typically, BFA was prepared according to a fermentation method. In a process, 5 g of the cleaned CS powder was placed into a Petri dish with the temperature maintained at 35 °C for 15 days, after which the resulting mixture was filtered, and the expected BFA was finally obtained after drying in an oven for 24 h.

### 2.3. Preparation of Methylated BFA (M-BFA)

In a typical process, 0.5 g of BFA was dissolved in 10 mL CH_3_OH with stirring for 10 min to obtain a uniform solution. Then, a homogeneous mixture was obtained after K_2_CO_3_ (0.5 g) and DMC (20 mL) were added into the solution with shocking for 30 min. After that, the mixture was transferred into a high-pressure reactor, followed by heating at 120 °C for 6 h and naturally cooling down to room temperature. Subsequently, the raw composite was obtained after drying overnight at 80 °C. Finally, the as-obtained composite was subjected to centrifugal treatment by using 20 mL of chloroform three times to remove the impurities, and the expected pure M-BFA was obtained after the supernatant was dried in a fume hood until the chloroform volatilized completely.

### 2.4. Preparation of Hydrogel

Hydrogel was prepared using M-BFA according to the methodology reported previously [25,26]; 0.5 g of the pure M-BFA was first mixed uniformly with 10 mL of deionized water in a 50 mL beaker under magnetic stirring. Then, 10 mL of DMAA (crosslinker) was added to the obtained mixture with continuous stirring at 40 °C for 10 min. After that, 0.2 g of MBA as a cocatalyst was added into the solution under magnetic stirring for another 5 min. Subsequently, 0.2 g of (NH_4_)_2_S_2_O_4_ as the initiator was added when the temperature reached 55 °C and was maintained for 2 h. Finally, the desired hydrogel was obtained after the mixture was freeze-dried for 24 h, and the obtained hydrogel was named M-BFA/DMAA and kept in a vacuum oven for further use. For comparison, another hydrogel was prepared without M-BFA under the same condition, and it was named DMAA. Figure 1 illustrates the specific preparation processes of M-BFA/DMAA.

### 2.5. Characteristics

Fourier transform infrared (FT-IR) spectra were obtained on a spectrometer (Shimadzu IR-Prestige-21, SHIMADZU, Chukyo ku, Kyoto, Japan) by employing dried KBr pressed powder discs. Scanning electron microscopy (SEM) was carried out using a field-emission microscope (JEOL JSM-6390LV, JEOL, Showa City, Tokyo) with an accelerating voltage of 15 kV. Measurements of nitrogen porosimetry were performed on an instrument (Quantachrome NOVA 1000e, Conta Instruments, Miami, FL, USA) with a temperature of −196 °C. The Brunauer–Emmett–Teller (BET) method was applied to acquire the specific surface area. In addition, the Barrett–Joyner–Halenda (BJH) method was used to calculate the pore volume and pore size distribution. X-ray photoelectron spectroscopy (Thermo escalab 250XI, Thermo Fisher Scientific, Waltham, MA, USA) was conducted to measure the chemical state compositions. A thermogravimetric instrument (Netzsch STA 449 F3 Jupiter, NETZSCH, Selber, Bavaria, Germany) with nitrogen as a carrier gas was used to assess the thermal stability of the samples, and data including the mass change and the initial and final temperatures of thermal decomposition were collected.

### 2.6. Tests of Swelling Property for Adsorbing Water

To determine the swelling property of the hydrogels, an equivalently sized sample (1 mg) was immersed in distilled water (1 mL). The hydrogels were removed from the water, and excess surface water was blotted with sterile tissue paper at various time intervals. In addition, the wet weights were tested at different time periods from 0 to 1500 min, different temperatures (25 °C, 35 °C, 45 °C, and 55 °C) and different pHs (1, 2, 3, 7, 10, 11 and 13) according to a sensitive balance. The percentage of the swelling ratio was calculated based on the following formula:Percentage of swelling (%) = [(W_w_ − W_d_)/W_d_] × 100(1)
in which W_w_ is the wet weight and W_d_ is the dry weight of the obtained samples.

## 3. Results and Discussion

The crosslinking structure of hydrogels plays an important role in the adsorbent performance. FT-IR spectra of BFA, M-BFA, DMAA and M-BFA/DMAA are shown in Figure 1a. The FT-IR spectra revealed that the peak located at 3420 cm^−1^ was mainly caused by the stretching vibration of -OH. Obviously, the -OH peak in B-MFA was weakened, which further indicated a reduced hydroxyl content in B-MFA, and the MFA was successfully methylated. Moreover, the M-BFA/DMAA also had a low hydroxyl content, which was due to the methylation of BFA reducing the -OH groups. Furthermore, in comparison with BFA, the characteristic absorption peak of -CH_2_- located at 1420 cm^−1^ in M-BFA was stronger, confirming that M-BFA was successfully methylated. It can be observed that BFA, M-BFA, DMAA and M-BFA/DMAA had C=O and C-O characteristic peaks of 1750 cm^−1^ and 1050 cm^−1^, respectively. These results indicated that M-BFA/DMAA was prepared successfully.

Typically, DMAA and M-BFA/DMAA were prepared through the vinyl monomer polymerization reaction with the release of heat because of the additional reaction of the double bond of the monomer. It is well known that the released heat is proportional to the enthalpy of polymerization, which was given by the integral of the area under the curve. The temperature values and enthalpy of polymerization of the vinyl monomer (HEMA) are presented in Figure 1b. It can be seen that the HEMAs of DMAA and M-BFA/DMAA were similar. More specifically, the two enlarged pictures confirm that the enthalpy change of M-BFA/DMAA was 221 J/g, which was much smaller than that of DMAA (245.8 J/g), confirming the successful preparation of M-BFA/DMAA, due to the formation of two single bonds during the methylation of BFA. In particular, the significant change in the thermogravimetric (TG/DTG) analyses is a powerful proof of polymer crosslinking. Notably, DTG curves (Figure 1c) confirmed that M-BFA/DMAA had a higher heat resistance than DMAA. Therefore, M-BFA/DMAA had better thermal stability due to the lower C=C concentration. Importantly, the thermal loss curve shows that it is greatly significant to manufacture M-BFA/DMAA on a large scale for agricultural applications. In addition, TG curves for DMAA and M-BFA/DMAA were obtained in a nitrogen atmosphere. From the TG curves in Figure 1d, it can be clearly seen that three main zones of weight loss are presented in DMAA and M-BFA/DMAA. In the first weight loss zone, the range of M-BFA/DMAA was from 0 to 381.5 °C, which was higher than that of DMAA (0–363.5 °C), confirming that M-BFA/DMAA had higher thermal stability than DMAA. The second and third weight loss zones of DMAA and M-BFA/DMAA could be related to the breaking of the polymer chains.

The chemical compositions of the DMAA and M-BFA/DMAA samples were analyzed by XPS. As shown in Figure 2a, three distinct peaks corresponding to O 1s, N 1s, C 1s in the full-width XPS spectrum were observed at 530.5 eV, 400.7 eV, 284.6 eV, confirming that both were composed of the elements O, N, and C, respectively. As shown in Figure 2b, two peaks at the binding energies of 532.0 and 530.6 eV were observed from the XPS of DMAA, which corresponded to C-O and C=O, respectively. In addition, the high-resolution O 1s spectrum of M-BFA/DMAA could be deconvoluted into two dominant peaks at 531.7 and 530.9 eV, corresponding to C=O and C-O, respectively. In comparison with DMAA, there was more C=O in M-BFA/DMAA, which was favorable for water adsorption. In XPS C 1s spectra (Figure 2c), the peaks at about 284.7 eV, 285.9 eV and 287.3 eV in DMAA could be attributed to C-C, C-O and C=O peaks, respectively; Meanwhile, the C 1s curve of M-BFA/DMAA also showed three characteristic peaks of C-C (284.5 eV), C-O (285.5 eV) and C=O (287.1 eV). In addition, the XPS N 1s spectra of M-BFA/DMAA and DMAA are displayed in Figure 2d for comparison. There were two distinguishable peaks in DMAA that appeared at 399.5 eV and 401.3 eV, corresponding to C-N and O=C-N. Different from DMAA, there was only one peak of C-N (399.4 eV) presented in M-BFA/DMAA, which was conducive to the adsorption of water molecules.

To study the specific surface area and pore size distribution of M-BFA/DMAA nanostructures at different power outputs, the adsorption–desorption isotherms of nitrogen and the corresponding pore size distributions for DMAA and M-BFA/DMAA are shown in Figure 3. It can be seen that DMAA and M-BFA/DMAA showed an IUPAC type IV isotherm (Figure 3a,c), indicating that they had a mesoporous structure. In addition, the BET surface area and pore volume of M-BFA/DMAA were 11.48 m^2^/g and 0.0068 m^3^/g, respectively, which were larger than those of DMAA (1.59 m^2^/g and 0.000575 m^3^/g). Therefore, it was stated that M-BFA/DMAA had a larger BET surface area and pore volume, so M-BFA/DMAA was able to offer more reactive sites for the adsorption process toward water than DMAA. Meanwhile, as presented in Figure 3b, the pore size distribution profile displayed that the pore width of M-BFA/DMAA was dominant in the range of 1.5–10.5 nm, confirming that M-BFA/DMAA had microporous and mesoporous structures. Different from M-BFA/DMAA, DMAA had a pore width of 2.0–10.0 nm (Figure 3d); thus, DMAA only had mesoporous structures. However, it was obviously observed that the mesoporous structure of M-BFA/DMAA was significantly more prominent than that of DMAA, which further confirmed that M-BFA/DMAA delivered more adsorption ability for water than DMAA due to the presence of a mesoporous structure favorable for the transfer of water.

The morphologies of DMAA and M-BFA/DMAA were investigated by scanning electron microscopy (SEM) techniques. As presented in Figure 4a,b, it could be clearly seen that DMAA exhibited a completely smooth surface; meanwhile, no porosity could be observed, which made DMAA deliver a poorer ability to adsorb and transfer water. Different from the morphology of DMAA, the obtained M-BFA/DMAA in the SEM observation displayed a rough surface area (see Figure 4c). In addition, the high-magnification image (Figure 4d) exhibited that M-BFA/DMAA had more pores than DMAA (Figure 4b). More importantly, the lightly porous structures were favorable for the easy accessibility of water molecules to the active sites in M-BFA/DMAA for water adsorption and storage. Therefore, it could be concluded that the swelling rate of M-BFA/DMAA was higher than that of DMAA. To verify this conclusion, their swelling rates were further tested.

To understand the swelling behaviors of the DMAA and M-BFA/DMAA, their swelling behaviors were investigated at different testing times, temperatures and pHs (as shown in Figure 5). As shown in Figure 5a, it was found that the time dependence of the swelling process consisted of two parts. Firstly, at the early stage of swelling, although both M-BFA/DMAA and DMAA displayed an increasing swelling rate before 500 min, the swelling rates of the M-BFA/DMAA specimen were 5.04, 7.48, 8.84 and 9.26 at 100, 200, 300, 400 and 500 min, respectively; these rates were much higher than those of pure DMAA at the same times (4.34, 6.91, 7.71, 8.30 and 8.49). Secondly, the swelling ratio of M-BFA/DMAA and DMAA remained stable after 500 min because the water adsorption reached saturation, but M-BFA/DMAA still had higher swelling ratios (9.19, 9.21, 9.26, and 9.23) than pure DMAA hydrogel (8.28, 8.29, 8.25 and 8.23) at the corresponding times of 1200, 1300, 1400 and 1500 min. Detailed values are shown in Figure 5b, in which it can be seen that M-BFA/DMAA gave a swelling rate of 10% higher than DMAA with the time ranging from 100 to 1500 min, especially at 300 min. Although their swelling rates tended to be stable after 500 min, the swelling rate of M-BFA/DMAA was still higher than that of DMAA of 0.9–1.0, which was consistent with the results shown in Figure 5a.

In addition, the temperature-dependent swelling behaviors of both DMAA and M-BFA/DMAA hydrogels in deionized water with temperatures ranging from 25 to 55 °C are displayed in Figure 5c. It can be clearly seen that the swelling ratios of M-BFA/DMAA were 9.10, 9.22, 8.10 and 8.03 at the temperatures of 25, 35, 45 and 55 °C, which were much higher than those of DMAA (8.34, 8.22, 7.79 and 7.22) at the same time. Accordingly, M-BFA/DMAA delivered a higher swelling ratio than DMAA. In addition, M-BFA/DMAA achieved the highest swelling ratio at 35 °C. In an assessment of their temperature sensitivity, both DMAA and M-BFA/DMAA displayed a temperature-responsive swelling behavior (Figure 5d). Clearly, M-BFA/DMAA displayed a temperature sensitivity of 0.16, similar to that of DMAA (0.1), because of the association and dissociation of hydrogen bonding within both DMAA and M-BFA/DMAA.

The pH value of the swelling medium has an important effect on the water absorption capacity of hydrogels. In order to observe the response of DMAA and M-BFA/DMAA at different pH conditions, they were allowed to swell to equilibrium with an aqueous medium at different pHs, and the effects of pH on their swelling behavior are summarized in Figure 5e. Clearly, M-BFA/DMAA delivered the swelling rates of 8.18, 8.82, 9.11, 9.23, 9.08, 8.69 and 8.09 at pH = 1, 2, 4, 7, 11, 12 and 13; meanwhile, the swelling rates of DMAA were 8.28, 8.34, 8.32, 8.23, 8.16, 8.22 and 8.19 at the same pH values, respectively. The results confirmed that although the swelling rates of M-BFA/DMAA were smaller than those of DMAA at pH = 1 and 13, its other swelling rates were much higher than those of DMAA at the corresponding pHs. More importantly, the swelling ratios of M-BFA/DMAA and DMAA were compared at the same pH, and the compared results were obtained according to the following equation: comparison of swelling ratio = Sw(M-BFA/DMAA)/Sw(DMAA), in which Sw(M-BFA/DMAA) and Sw(DMAA) represent the swelling ratios of M-BFA/DMAA and DMAA at the different pHs, respectively. The corresponding results are shown in Figure 5f; it can be seen that the comparison of swelling ratios were 0.99, 1.06, 1.09, 1.12, 1.11, 1.06 and 0.99 at the corresponding pH = 1, 2, 4, 7, 11, 12 and 13, which further confirmed that M-BFA/DMAA delivered higher swelling rates than DMAA at pH = 2, 4, 7, 11, and 12, but not at pH values of 1 and 13. Therefore, it could be obviously seen that the obtained M-BFA/DMAA hydrogel swelled the most in neutral conditions in comparison with acidic pH and basic pH, confirming it was suitable for practical applications.

To study swelling retention of as-obtained DMAA and M-BFA/DMAA, they were immersed in a deionized water solution under the same conditions. Figure 6a,b display the photographs of such DMAA and M-BFA/DMAA samples before and after swelling in deionized water at 328 K and pH = 7. It could be clearly found that the swollen DMAA and M-BFA/DMAA were still able to maintain their shapes. Amazingly, the shape of the swollen M-BFA/DMAA was larger than that of the swollen DMAA, which suggested that the swollen M-BFA/DMAA had a superior swelling performance in comparison to the swollen DMAA because M-BFA/DMAA had the new crosslinked structure.

In order to further investigate their application performance, their anti-swelling properties were collected, as displayed in Figure 6c. It could be clearly seen that DMAA and BFA/DMAA showed the same anti-swelling performance at the initial time. With the increase in time, their anti-swelling properties decreased. However, BFA/DMAA displayed a better anti-swelling performance because the anti-swelling property of BFA/DMAA was higher than that of DMAA in the whole process. In particular, the anti-swelling property of BFA/DMAA was greater than that of DMAA.

To further assess their performances, their water loss rate was also analyzed according to Figure 6c, and the obtained results are displayed in Figure 6d. According to Equation (1), the water loss rates of the BFA/DMAA were found to be 7.43%, 24.70%, 27.71%, 30.46%, 32.75 and 35.17% at 30, 60, 90, 120, 150 and 200 min, respectively, which were higher than those of DMAA (5%, 9%, 13%, 15%, 17% and 19%) at the same times. The enhanced performance of BFA/DMAA should be ascribed to hydrophilic groups, which greatly improved its adsorbent ability for water.

## 4. Conclusions

In this work, M-BFA/DMAA hybrid hydrogels were successfully prepared by combining M-BFA and DMAA via the oxidation and freezing method. The XPS results revealed that there were more hydrophilic groups (C=O and C=N) in M-BFA/DMAA than in DMAA, which was favorable for water adsorption. In addition, the BET and SEM confirmed M-BFA/DMAA as a porous material with a higher surface area and pore volume than DMAA, which was conducive to the water transfer and helpful for water molecule accessibility to active sites, boosting its water adsorption and storage ability. It was found that the swelling behaviors of the M-BFA/DMAA hydrogel were 0.77 higher than those of DMAA at 500 min, pH = 7 and 35 °C, confirming that the as-prepared M-BFA/DMAA hydrogel as a water-retaining agent had a better performance towards water adsorption and storage. The influence of M-BFA/DMAA on swelling behavior was related to the HA’s methylation, which was favorable for the preparation of hydrogels with polar groups to keep the good swelling capability. All the results illustrated that the as-prepared M-BFA/DMAA hydrogel possessed high water absorption, water retention properties and easy biodegradation; thus, it could be applied in ecological restoration as well as soil and water conservation projects in arid and semi-arid areas. This work shows that M-BFA/DMAA can be an ideal adsorbent candidate for applications in arid and semi-arid regions.

## Data Availability

Data are contained within the article.

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
