# Peer review of "Methylated Biochemical Fulvic Acid-Derived Hydrogels with Improved Swelling Behavior and Water Retention Capacity"

_materials, 2024, doi:10.3390/ma17061448_

Round 1

Reviewer 1 Report

Comments and Suggestions for Authors

The MS “Methylated Biochemical Fulvic Acid Derived Hydrogels with 2 Improved Swelling Behavior and Water Retention Capacity” by Shi et.al., addresses an important real-world problem of water shortage and potential application of hydrogels as water retaining agents using natural and biodegradable humic acid is a good approach. Swelling experiments under different conditions provide good insights into water absorption and retention abilities - key parameters for practical use. The methylated hydrogel demonstrates consistently higher swelling across time points, pH and temperature. 

I have some minor comments which need to address.

1.      Please rewrite experimental section 2.3 and 2.4 with more scientific way.

2.      In Fig 1, figure legend missing b, c, d details

3.      In page 7, line 241, please correct Fig 5a not Fig 4a

4.      In Fig 5f, the comparison should be done at same pH and please include it in results and discussion.

5.      Statistical analysis is missing in presenting the measured data which have some variability. Error bars should be added.

 Thanks

Comments on the Quality of English Language

rewrite section 2.3 and 2.4. 

Author Response

The MS “Methylated Biochemical Fulvic Acid Derived Hydrogels with  Improved Swelling Behavior and Water Retention Capacity” by Shi et.al., addresses an important real-world problem of water shortage and potential application of hydrogels as water retaining agents using natural and biodegradable humic acid is a good approach. Swelling experiments under different conditions provide good insights into water absorption and retention abilities - key parameters for practical use. The methylated hydrogel demonstrates consistently higher swelling across time points, pH and temperature. 

I have some minor comments which need to address.

  1. Please rewrite experimental section 2.3 and 2.4 with more scientific way.

Response: Thank you very much for the reviewer`s comment. We carefully think of our experimental in details and rewrite the experimental section 2.3 and 2.4 in our manuscript again, as depicted in the following:

Experimental 2.3: In a typical process, 0.5 g of BFA was dissolved in 10 mL of CH3OH with stirring for 10 min to gain a uniform solution. Then, a homogeneous mixture was obtained after K2CO3 (0.5 g) and DMC (20 mL) were added into the solution with shocking for 30 min. After that, the mixture was transferred into a high-pressure reactor, followed by heating at 120 oC for 6 h and naturally cooling down to room temperature. Subsequently, the raw composite was obtained after drying overnight at 80 oC. Finally, the as-obtained composite was centrifugal treatment by using 20 mL of chloroform with three times to remove the impurities, and the expected pure M-BFA was obtained after the supernatant was dried in fume hood until chloroform volatilized completely.

Experimental 2.4: Typically, 0.5 g of the pure M-BFA was firstly mixed uniformly with 10 mL of deionized water in a 50 mL beaker under magnetic stirring. Then, 10 mL of DMAA (crosslinker) was added into the obtained mixture with continuous stirring at 40 oC for 10 min. After that, 0.2 g of MBA as a cocatalyst was added into the solution under magnetic stirring for another 5 min. Subsequently, 0.2 g of (NH4)2S2O4 as the initiator was added with the temperature reaching up to 55 oC  maintaining for 2 h. Finally, the desired hydrogel was obtained after the mixture was freeze-dried for 24 hours, and the obtained hydrogel was named as M-BFA/DMAA and kept in a vaccum oven for further use. For comparison, another hydrogel was prepared without M-BFA under the same condition, and it was named as DMAA.

  1. In Fig 1, figure legend missing b, c, d details

Response: Thanks very much for the reviewer`s comment. We added the missing b, c, d details in our manuscript carefully.

Figure 1. (a) FT-IR spectra of BFA, M-BFA, DMAA, and M-BFA/DMAA; (b) DSC plots, (c) DTA curves and (d) TG curves of DMAA and M-BFA/DMAA, respectively

  1. In page 7, line 241, please correct Fig 5a not Fig 4a

Response: Thank you very much for the review`s comment. As required, we correct Fig 5a in stand of Fig 4a in our manuscript.

Although their swelling rates tend to be stable after 500 min, the swelling rate of M-BFA/DMAA were still higher than that of DMAA of 0.9-1.0, which was consistent    with the results in Figure 5a.   

  1. In Fig 5f, the comparison should be done at same pH and please include it in results and discussion.

Response: Thank you very much for the review`s suggested comment. We compared the swelling ratio of M-BFA/DMAA and DMAA at the same pH and included it in results and discussion, as displayed in the following:

Figure 5. (f) Comparison of swelling ratio of M-BFA/DMAA and DMAA at the same pH

More importantly, in order to clearly compare the swelling ratio of M-BFA/DMAA and DMAA at the same pH, the compared results were obtained according to the equation: comparison of swelling ratio=Sw(M-BFA/DMAA)/Sw(DMAA), in which Sw(M-BFA/DMAA) and Sw(DMAA) represented the swelling ratio at the different pHs. The corresponding results were shown in Figure 5f, it could be seen that the comparison of swelling ratio was 0.99, 1.06, 1.09, 1.12, 1.11, 1.06 and 0.99 at the corresponding pH=1, 2, 4, 7, 11, 12, and 13, which further confirming that M-BFA/DMAA delivered the higher swelling rates that of DMAA at the corresponding pH=2, 4, 7, 11, and 12, except pH were 1 and 13.  

  1. Statistical analysis is missing in presenting the measured data which have some variability. Error bars should be added.

Response: Thank you very much for the review`s good comment. We carefully verify our manuscript, and there were two pictures missing Error bars. Therefore, they were added in the corresponding pictures, as shown in the following:

Figure 5. The swelling behaviors of DMAA and M-BFA/DMAA at different testing times (a) 

Figure 6. Anti-swelling property (c) of DMAA and BFA/DMAA

Reviewer 2 Report

Comments and Suggestions for Authors

The paper describes the synthesis of methylated fulvic acid-based hydrogels and their swelling behavior and water retention capacity.

Some changes must be made to improve the quality of the manuscript and make it worthy of publication in Materials journal.

1. I suggest to the authors the introduction of the synthetic route of the hydrogel instead of Scheme 1.

2. The purification of M-BFA is not described (lines 95-100).

3. Where are b, c, d in the title of Figure 1?

3. I suggest to the authors to introduce more comments about the decomposition stages and XPS spectra of M-BFA/DMAA and DMAA.

4. The authors must indicate the value of “low- and high” magnification images of M-BFA/DMAA and DMAA (Figure 4)

Author Response

Title: Methylated Biochemical Fulvic Acid Derived Hydrogels with Improved Swelling Behavior and Water Retention Capacity

Manuscript ID: materials-2764951

Keywords: Humic acid; Hydrogel; Swelling rate; Water-retaining agent

The paper describes the synthesis of methylated fulvic acid-based hydrogels and their swelling behavior and water retention capacity. Some changes must be made to improve the quality of the manuscript and make it worthy of publication in Materials journal.

  1. I suggest to the authors the introduction of the synthetic route of the hydrogel instead of Scheme 1.

Response: Thank you very much for the reviewer`s suggested comment. We carefully corrected the synthetic route of the hydrogel and changed it, as shown in Scheme.1.

Scheme 1. Schematic the synthetic route of M-BFA/DMAA.

  1. The purification of M-BFA is not described (lines 95-100).

Response: Thank you very much for the reviewer`s good comment. Because the as-prepared M-BFA was dissolved in organic solution, chloroform was used to obtain pure M-BFA. Therefore, the purification of M-BFA was added and described in Experimental 2.3, as the details shown in the following:

Experimental 2.3: In a typical process, 0.5 g of BFA was dissolved in 10 mL of CH3OH under magnetic stirring for 10 min to gain a uniform solution. Then, a homogeneous mixture was obtained after K2CO3 (0.5 g) and DMC (20 mL) were added into the solution with ultrasonic treatment for 30 min. After that, the mixture was transferred into a high-pressure reactor, followed by heating at 120 oC for 6 h and naturally cooling down to room temperature. Subsequently, the raw composite was obtained after drying overnight at 80 oC. Finally, the as-obtained composite was centrifugal treatment by using 20 mL of chloroform with three times to remove the impurities, and the expected pure M-BFA was obtained after the supernatant was dried in fume hood until chloroform volatilized completely.

  1. Where are b, c, d in the title of Figure 1?

Response: Thanks very much for the reviewer`s comment. We added the missing b, c, d details in our manuscript carefully.

Figure 1. (a) FT-IR spectra of BFA, M-BFA, DMAA, and M-BFA/DMAA; (b) DSC plots, (c) DTA curves and (d) TG curves of DMAA and M-BFA/DMAA, respectively

  1. I suggest to the authors to introduce more comments about the decomposition stages and XPS spectra of M-BFA/DMAA and DMAA.
  2. Response: Thanks very much for the reviewer`s suggested comment. We introduced more comments about the the decomposition stages and XPS spectra of M-BFA/DMAA and DMAA as following:

As shown in Figure 2b, two peaks at binding energies of 532.0 and 530.6 eV were observed from the XPS of DMAA, which corresponded with C-O and C=O, respectively. Besides, The high-resolution O 1s spectrum of M-BFA/DMAA could be deconvoluted into two dominant peaks at 531.7 and 530.9 eV, corresponding to C=O and C-O, respectively. In particular, there was more C=O in M-BFA/DMAA, which was favorable for water adsorption. In XPS C 1s spectra (Figure 2c), the peaks at about 284.7 eV, 285.9 eV and 287.3 eV in DMAA could be attributed to C-C, C-O and C=O peaks, respectively; Meanwhile, the C 1s curve of M-BFA/DMAA also showed three characteristic peaks of C-C (284.5 eV), C-O (285.5 eV) and C=O (287.1 eV). In addition, the XPS N 1s spectra of M-BFA/DMAA and DMAA were displayed in Figure 2d for comparison. There were two distinguishable peaks in DMAA appeared at 399.5 eV and 401.3 eV , corresponding to C-N and O=C-N. However, there was only one peak of C-N (399.4 eV) presenting in M-BFA/DMAA. It was worth noting that the presence of C-N was conducive to the adsorption of water molecules.

  1. The authors must indicate the value of “low- and high” magnification images of M-BFA/DMAA and DMAA (Figure 4)

Response: Thanks very much for the reviewer`s suggested comment. We carefully see all of the low- and high magnification images of M-BFA/DMAA and DMAA in our manuscript, and strictly added the corresponding value for each image in Figure 4, as the images shown in the following:

Figure 4. Low-magnification image of (a) DMAA and (c) M-BFA/DMAA, respectively;High-magnification image of (b) DMAA and (d) M-BFA/DMAA, respectively.

Reviewer 3 Report

Comments and Suggestions for Authors

This manuscript entitled "Methylated biochemical fulvic acid derived hydrogels with improved swelling behavior and water retention capacity" reports interesting, and I think novel, data.

However, it seem to me that authors hurried to write their manuscript. A number of leaving out important information, prevent the publication of this manuscript in its present version.

In more details:

(a) Section / Introduction, line 69: Authors should correct [23,42] to [23,24],

(b)  Section / Experimental: In sub-sections Preparation of BFA, Preparation of Methylated BFA (M-BFA), and Preparation of Hydrogel: Authors, altough give a detailed description of the followed methods, however give no references; the reader is confused. Authorst they either discovered all the three aforementioned methods or they found them in the literature. In the former case they should point out the fact, whereas in the latter case they should cite the corresponding refereces.

(c) The abovementioned "logic" runs through the section "Results and Discussion" as well, and it should be faced positevely by authors, as it is suggested in the comments, which are mentioned just above.

(d) Especially in the section "Conclusions", the reader suddenly reads comparisons but cannot find to whom/what the findings of the manuscript are compared. I am referred to:

   (i) Lines 298 to 301: Comparison without citation,

   (ii) Lines 301 to 304: Ditto, and

  (iii) Lines 304 to 307: Why the higher swelling behaviors of the M-BFA/DMAA hydrogel 0.77, 1.00 and 1.00 are significant towards water adsorption and storage? The corresponding cItations should be given, and the comparisons should be described in details, in all the (i), (ii) and (iii) cases.

Overall: I suggest to authors a minor revision, i.e., to read carefully their manuscript, prepare a revised version of it according to my comments, and resubmit it for consideration. The manuscript cannot be published in the Journal "Materials", in its present version.

Comments on the Quality of English Language

A minor editing of English language is required.

Author Response

Title: Methylated Biochemical Fulvic Acid Derived Hydrogels with Improved Swelling Behavior and Water Retention Capacity

Manuscript ID: materials-2764951

Keywords: Humic acid; Hydrogel; Swelling rate; Water-retaining agent

This manuscript entitled "Methylated biochemical fulvic acid derived hydrogels with improved swelling behavior and water retention capacity" reports interesting, and I think novel, data.

However, it seem to me that authors hurried to write their manuscript. A number of leaving out important information, prevent the publication of this manuscript in its present version.

In more details:

  • Section/Introduction, line 69: Authors should correct [23,42] to [23,24],

Response: Thanks very much for the reviewer`s comment. We carefully verified the manuscript and have corrected [23,42] to [23,24], as shown in the following:

Unfortunately, the practical application of the water-retaining materials prepared with HAs is severely limited due to their low solubility in organic solvents [23,24].

  • Section/Experimental: In sub-sections Preparation of BFA, Preparation of Methylated BFA (M-BFA), and Preparation of Hydrogel: Authors, altough give a detailed description of the followed methods, however give no references; the reader is confused. Authorst they either discovered all the three aforementioned methods or they found them in the literature. In the former case they should point out the fact, whereas in the latter case they should cite the corresponding referen

Response: Thanks very much for the reviewer`s comment. In order to point out the fact and help readers understand our manuscript very well, We strictly find out the suitable references for our manuscript, and cited them as following:

  1. Sun, X. F.; Wang, S. G.; Liu, X. W.; Liu, X.W.; Gong, W.X.; Bao, N.; Ma, Y.;The effects of pH and ionic strength on fulvic acid uptake by chitosan hydrogel beads.Colloid Surface A 2008, 324, 28-34.
  2. Sirousazar, M.; Khodamoradi, P.; Freeze-thawed humic acid/polyvinyl alcohol supramolecular hydrogels. Mater. Today Commun. 2020, 22, 100719.
  • The abovementioned "logic" runs through the section "Results and Discussion" as well, and it should be faced positively by authors, as it is suggested in the comments, which are mentioned just above.

Response: Thanks very much for the reviewer`s comment. We carefully read the the section "Results and Discussion" again and again; Meanwhile, we faced positively it and corrected some sentences in our manuscript strictly.

(d) Especially in the section "Conclusions", the reader suddenly reads comparisons but cannot find to whom/what the findings of the manuscript are compared. I am referred to:

 (i) Lines 298 to 301: Comparison without citation,

Response: Thanks very much for the reviewer`s comment. We carefully read our manuscript and comparison of M-BFA/DMAA and DMAA. So the sentence was corrected as following:

The XPS results revealed that there were more hydrophilic groups (C=O and C=N) in M-BFA/DMAA than those in DMAA, which was favorable for water adsorption. Besides, the BET and SEM confirmed M-BFA/DMAA as a porous material with a higher surface area and pore volume than DMAA, which was conducive to the water transfer and helpful for water molecule accessible to active sites to boost its ability of water adsorption and storage.

 (ii) Lines 301 to 304: Ditto, and

Response: Thanks very much for the reviewer`s good comment. The surface area and pore volume of M-BFA/DMAA and DMAA were compared carefully, and the sentence was corrected as following:

Besides, the BET and SEM confirmed M-BFA/DMAA as a porous material with a higher surface area and pore volume than DMAA, which was conducive to the water transfer and helpful for water molecule accessible to active sites to boost its ability of water adsorption and storage.

(iii) Lines 304 to 307: Why the higher swelling behaviors of the M-BFA/DMAA hydrogel 0.77, 1.00 and 1.00 are significant towards water adsorption and storage?

Response: Thanks very much for the reviewer`s comment. We have carefully verified our manuscript again and again, and strictly confirmed the experimental results. The sentence was corrected as following:

It was found that the swelling behaviors of the M-BFA/DMAA hydrogel were 0.77 higher than those of DMAA at 500 min, pH=7 and 35 oC, confirming that the as-prepared M-BFA/DMAA hydrogel as a water-retaining agent had a better performance towards water adsorption and storage.

The reason for the M-BFA/DMAA hydrogel had the higher swelling behaviors was shown as following: The M-BFA/DMAA hydrogel had more hydrophilic groups (C=O and C=N) favorable for water adsorption. Besides, the M-BFA/DMAA hydrogel had a porous structure with a higher surface area and pore volume, which was conducive to the water transfer and helpful for water molecule accessible to active sites to boost its ability of water adsorption and storage.

The corresponding citations should be given, and the comparisons should be described in details, in all the (i), (ii) and (iii) cases.

Overall: I suggest to authors a minor revision, i.e., to read carefully their manuscript, prepare a revised version of it according to my comments, and resubmit it for consideration. The manuscript cannot be published in the Journal "Materials", in its present version.
